# An Arctic Ozone Hole in 2020 If Not For the Montreal Protocol

Catherine Wilka[1], Susan Solomon[1], Doug Kinnison[2], David Tarasick[3]

[1]Department of Earth, Atmospheric and Planetary Sciences, Massachusetts Institute of Technology, Cambridge, MA, USA
[2]Atmospheric Chemistry Division, National Center for Atmospheric Research, Boulder, CO, USA
[3]Environment and Climate Change Canada, Toronto, ON, Canada

*Correspondence to*: Catherine Wilka (cwilka@mit.edu)

**Abstract.** Without the Montreal Protocol the already extreme Arctic ozone losses in boreal spring of 2020 would be expected to have produced an Antarctic-like ozone hole, based upon simulations performed using the Specified Dynamics version of the Whole Atmosphere Community Climate Model (SD-WACCM) using an alternate emission scenario of 3.5% growth in ozone depleting substances from 1985 onwards. In particular, we find that the area of total ozone below 220 DU, a standard metric of Antarctic ozone hole size, would have covered about 20 million $km^2$. Record observed local lows of 0.1 ppmv at some altitudes in the lower stratosphere seen by ozonesondes in March 2020 would have reached 0.01, again similar to the Antarctic. Spring ozone depletion would have begun earlier and lasted longer without the Montreal Protocol, and by 2020 the year-round ozone depletion would have begun to dramatically diverge from the observed case. This extreme year also provides an opportunity to test parameterizations of polar stratospheric cloud impacts on denitrification, and thereby to improve stratospheric models of both the real world and alternate scenarios. In particular, we find that decreasing the parameterized nitric acid trihydrate number density in SD-WACCM, which subsequently increases denitrification, improves the agreement with observations for both nitric acid and ozone. This study reinforces that the historically extreme 2020 Arctic ozone depletion is not cause for concern over the Montreal Protocol's effectiveness, but rather demonstrates that the Montreal Protocol indeed merits celebration for avoiding an Arctic ozone hole.

## 1 Introduction

In the 1970s, Molina and Rowland issued a prescient warning to humanity that the chlorofluorocarbons (CFCs) contained in popular refrigerants, building foams, and aerosol cans posed a danger to the stratospheric ozone layer (Molina and Rowland, 1974). This threat, initially thought to be a worry for the next century, suddenly transformed into a pressing concern with the discovery of unexpected, deep springtime depletion in the Antarctic polar vortex (Farman et al., 1985), which became known to the world as the "ozone hole." Subsequent work (Solomon et al., 1986) revealed that heterogeneous chemical reactions involving chlorine and bromine on the cold surfaces of Polar Stratospheric Clouds (PSCs) were the missing link in the sequence of steps leading to this deep depletion. PSCs are made of water ice, nitric acid trihydrate (NAT), or supercooled ternary solutions of water, nitric acid, and sulfur (STS), and several studies have highlighted a significant role for sedimentation of large NAT particles in the removal of $HNO_3$ from the lower stratosphere, or denitrification (Toon et al., 1986; Crutzen and Arnold, 1986). Definitions for what constitutes an ozone hole have been debated in the scientific literature (see Langematz et al., 2018 and references therein) but for purposes of comparison to the discovery of the Antarctic ozone hole and its impact on policy of the era, here we use the historical definition of total ozone area below 220 Dobson Units (Stolarksi et al., 1990). Another important metric is extreme locally depleted ozone mixing ratios in the lower stratosphere (Hofmann et al., 1997), providing an important fingerprint for chemical ozone loss driven by chlorine chemistry on PSCs. In response to the increasing ozone depletion, the global community came together to pass the 1987 Montreal Protocol on Substances that Deplete the Ozone Layer, more commonly referred to as the Montreal Protocol. The Montreal Protocol, and its subsequent amendments during the 1990s, mandated the decrease and eventual cessation of worldwide production of ozone depleting substances (ODSs) such as CFCs (Birmpili, 2018).

Within the past few years, ever-stronger evidence for global ozone stabilization and a nascent Antarctic ozone recovery has emerged (Solomon et al., 2016; Sofieva et al., 2017; Chipperfield et al., 2017; Strahan and Douglass 2018). Despite uncertainties surrounding continuing CFC emissions from both scattered rogue production (Montzka et al., 2018) and existing stores in building foams and other banks (Lickley et al., 2020), the world appears to be on track for near-complete ozone recovery to near 1980s values as a result of decreasing ODSs by the second half of the 21st century (WMO 2018), and the Protocol has been ratified by every state represented at the United Nations. No other global environmental treaty can claim such a resounding success.

Success of the Montreal Protocol, however, should be measured not just by emission adherence but by the harm to the earth system and human society that its passage avoided. The first study on such a "World Avoided" (WA) by (Prather et al., 1996) found strong evidence that the Antarctic ozone hole would have continued to worsen on average. Subsequent studies

broadened to examine variability from year to year and at various longitudes. A decade later, (Morgenstern et al., 2008) studied WA in a more detailed three-dimensional model and found significant ozone decline in the upper stratosphere and polar vortices, a transition in the Arctic from dynamical to chemical control of ozone evolution, and major regional climate impacts caused by dynamical changes. The first fully interactive time-evolving global study of the world avoided by the Montreal Protocol, by (Newman et al., 2009), found increasingly extreme impacts throughout the 21$^{st}$ century. Their simulations for the WA predicted Arctic column ozone levels of 220 DU or less by 2030, with some minimum values within the vortex that low by 2020 in extreme cold years. The associated column depletion was predicted to yield a 550% increase in DNA damage when compared to 1980 by 2065 for NH midlatitudes. Chipperfield et al. (2015) examined the WA for the recent extreme cold year of 2011. They found that Arctic ozone levels would indeed have dropped below 220 DU in that year in the WA, but in a limited region that did not span the entire pole as in the Antarctic, as we discuss in Section 3. During a recent study quantifying drivers of depletion in Spring 2020 in the real world (Feng et al., 2021), they updated their previous WA model and found that the year 2020 would have seen much deeper column depletion during March than 2011 did (see their Supplementary Information).

The Arctic spring of 2020 displayed very cold temperatures and a stable polar vortex that led to record levels of Arctic PSCs, and deep Arctic ozone depletion in the real world at some altitudes in the lower stratosphere, as has already been shown using both ozonesondes (Wohltmann et al., 2020) and satellites (Manney et al., 2020). While the classical definition of an ozone hole (a significant areal extent below 220 DU) did not occur in 2020, many news reports characterized it as such, sparking public uncertainty over whether humanity has really solved the problem of ozone depletion. Here we seek to examine the chemistry and ozone depletion of both the real world and of that obtained in a world without the Montreal Protocol to evaluate what 2020 implies for the Montreal Protocol's achievements in the context of Arctic ozone loss.

## 2 Methods and Data

### 2.1 Model

We use the Specified Dynamics version of NCAR's Whole Atmosphere Community Climate Model (SD-WACCM) to compare the ozone depletion and chemistry in a simulation of the real world (RW) to one in which ODSs continued to increase at 3.5% per year from the year 1985 onwards (WA). The assumption of 3.5% per year growth matches that used in the Garcia et al. (2012) World Avoided study and is a good approximation of the grown rates seen in years immediately prior to emissions controls, thus representing an illustrative "business as usual" alternate trajectory. The Community Earth System Model, version 2 (CESM2) WACCM is a superset of the Community Atmosphere Model, version 6 (CAM6), which extends from the Earth's surface to the lower thermosphere (Gettelman et al., 2019). WACCM includes updated representations of boundary layer processes, shallow convection and liquid cloud macrophysics, and two-moment

cloud microphysics with prognostic cloud mass and concentration (Danabasoglu et al., 2020). Aerosol representation for dust, sea-salt black carbon, organic carbon, and sulfate in three size categories is prognostic in this version (Mills et al., 2016). We use the specified dynamics (SD) version of WACCM, where the atmosphere below 50 km is nudged to the Modern-Era Retrospective Analysis for Research and Applications version 2 (MERRA-2; Gelaro et al., 2017) temperature and wind fields with a relaxation time of 50 hours. There are 88 vertical pressure grid levels from the ground to the thermosphere (~140 km), with the altitude resolution increasing from ~0.1 km near the surface to ~1.0 km in the upper troposphere-lower stratosphere (UTLS) and ~1-2 km in the stratosphere. The horizontal resolution is 1.95° x 2.5° in latitude and longitude. All model results are taken from a 24-hour average for each given day. The chemistry mechanism used in this study includes a detailed representation of the middle atmosphere, with a sophisticated suite of gas-phase and heterogeneous chemistry reactions including the $O_x$, $NO_x$, $HO_x$, $ClO_x$, and $BrO_x$ reaction families (Kinnison et al., 2007). There are ~100 chemical species and ~300 chemical reactions. Reaction rates are updated following JPL 2015 recommendations (Burkholder et al., 2015). The model's volcanic sulfur loading is from the Neely and Schmidt (2016) database and has been updated through the Raiakoke eruption in 2019. Polar Stratospheric Clouds (PSCs) are present below ~200 K as solid nitric acid trihydrate (NAT), water ice, and super-cooled ternary solutions (Solomon et al., 2015). As described further in Sec 3, to simulate ozone loss more accurately we tested multiple values of the parameterized NAT particle number density controlling denitrification in this model ranging from the default of 0.01 particles per $cm^3$ to $10^{-5}$ particles per $cm^3$ and chose the smallest value for the final RW and WA simulations.

The Real World runs use the Coupled Model Intercomparison Project phase 6 (CMIP6) hindcast scenario (Meinshausen et al., 2017) based on observations for the evolution of ODSs and other emissions through 2014. The period 2015 through April 2020 uses the CMIP6 Shared Socioeconomic Pathways (SSP) 585 projection (O'Neill et al., 2016). The WA run assumes a 3.5% per year increase beginning in 1985 in all organic chlorine and bromine species, except for $CH_3Cl$, $CH_2Br_2$, and $CHBr_3$, which mainly have natural sources (Figure S1). $CH_3Br$ is assumed to be half from natural sources, half from anthropogenic, so that its increase is half that of the other ODSs.

## 2.2 Satellites

We compare SD-WACCM's total column ozone values to those observed by NASA's Ozone Monitoring Instrument (OMI) on the Earth Observing System (EOS) Aura satellite (Bhartia, 2012). OMI is a nadir-viewing wide-field-imaging spectrometer that continues the global total column ozone record from NASA's Total Ozone Mapping Spectrometer (TOMS). We use the Level 3 gridded data product here for comparison with SD-WACCM's daily column ozone values. Level 3 data is generated from high-quality only Level 2 data and is available on a daily basis. When calculating daily polar cap minimum total ozone values we filter data at solar zenith angles above 82° to remove spurious points.

We compare SD-WACCM's $HNO_3$ mixing ratios to those observed by NASA's Microwave Limb Sounder (MLS) on the EOS Aura satellite (Waters et al., 2006; Lambert et al., 2007). MLS has been continuously observing the upper atmosphere since its launch in 2004, although data gaps exist, including during the second half of March through early April 2020. MLS data was processed according to the flags and thresholds described in the Version 4.2x Level 2 Data Quality and Description Document. The vertical resolution of the $HNO_3$ data at the levels of interest is 3-4 km, with a reported measurement precision of ±0.6 ppbv and a systematic uncertainty of ±1.0 ppbv. MLS data was binned into a 5°x5° latitude-longitude grid before plotting.

## 2.3 Ozonesondes

We use balloon-based ozonesondes to examine ozone mixing ratios at individual levels and in vertical profiles. Ozonesondes are launched at regular intervals from multiple stations across the globe and collated by the World Ozone and Ultraviolet Data Centre. We use data from Resolute (74.86°N, -94.98°E), Ny-Ålesund (78.93°N, 11.88°E), Sodankyla (67.34°N, 26.51°E), Eureka (80.04°N, -86.18°E), Alert (82.49°N, -62.42°E), Lerwick (60.13°N, -1.18°E), and Thule (76.53°N, -68.74°E) to represent the historical record of ozone mixing ratios at 50 mb. All data in recent decades are from electrochemical concentration cell (ECC) ozonesondes, which have a precision of 3-5% and an overall uncertainty in ozone concentration of about ±10% up to 30 km (Smit et al., 2007; Tarasick et al., 2021). The ozone sensor response time of ~25 s, for a typical balloon ascent rate of 4-5 m/s, gives ozonesondes a vertical resolution of about 100-150 meters. Pre-1980 data from Resolute are from older Brewer-Mast sondes, which have a precision of about 5-10% (Kerr et al., 1994; Smit et al., 1998). For profile comparisons we use ECC ozonesondings from Eureka, Alert, and Resolute, along with simultaneously measured temperatures.

## 3 Results

Figure 1 shows the total column ozone (TCO) that would have been expected in the World Avoided (top left) on the day of greatest ozone depletion in 2020 in the model, March 13[th], compared to that expected and observed in the RW run (top right, bottom right) for the same day. The area meeting the standard definition of an ozone hole in the WA is nearly 20 million km², a comparable areal extent to many observed past Antarctic ozone holes, and the region below 150 DU stretches across the North Pole and over significant parts of Canada, Greenland, and Russia. By comparison, while the depletion in the RW case is clearly visible, it never breaches the 220 DU threshold for any significant area. Observations from OMI (lower right) support this, although the higher resolution satellite finds small, isolated patches below 220 DU. The difference between the WA and RW runs (lower left) is 20 DU or more throughout the Northern Hemisphere and maximizes at over 130 DU in the Arctic. We can compare this to another recent cold year, 2011 (Figure S2), previously highlighted by others for its large Arctic ozone losses in a WA simulation (Chipperfield et al. (2015), compare our Figure S2 with their Figure 3;

we note that our study follows a slightly different WA emissions path, with a different partitioning between anthropogenic and natural emissions for $CH_3Br$ in particular, compare our Figure S1 with their Figure 1a). In our simulations, the expected Arctic ozone hole in 2011 is much smaller in area than in 2020 (11.08 million $km^2$ vs 19.71 million $km^2$). The difference is partly due to the increased chlorine loading in the WA nine years later, but we also note that, while 2011 was an extremely cold year, 2020 had lower minimum ozone values that lasted longer than in 2011, with record fractions of the polar vortex being below the PSC temperature threshold for a longer period of time (Wohltmann et al., 2020; Inness et al., 2020). In summary, without the Montreal Protocol, the 2020's combination of extreme meteorology and increased chlorine loading would have resulted in unprecedented Arctic ozone depletion and an Arctic ozone hole comparable in areal extent to those of the Antarctic, with accompanying large impacts on UV levels throughout the Arctic.

To confirm the historically anomalous nature of 2020 and to evaluate our model's performance in more detail, we examine a time series of measurements at the 50 mb pressure level from archived Arctic ozonesondes, shown in the left panel of Figure 2. Because 2020 displayed very large local changes in Arctic ozone, the less-precise measurements from older Brewer-Mast sondes are also valuable for this purpose and are shown with open symbols. The long ozonesonde record allows us to compare to historical values predating the start of the satellite era in 1979, which is especially important for ozone trends as there may have been some depletion already by that time. Figure 2 shows ozone values for available days in March, stacked by year, and demonstrates that 2020 displays ozone amounts lower than any other year in the record at this altitude (including 2011, which displays the next deepest depletion). This is especially apparent in the log-scale version in Figure S3.

# Total Column O$_3$ on 13-Mar-2020

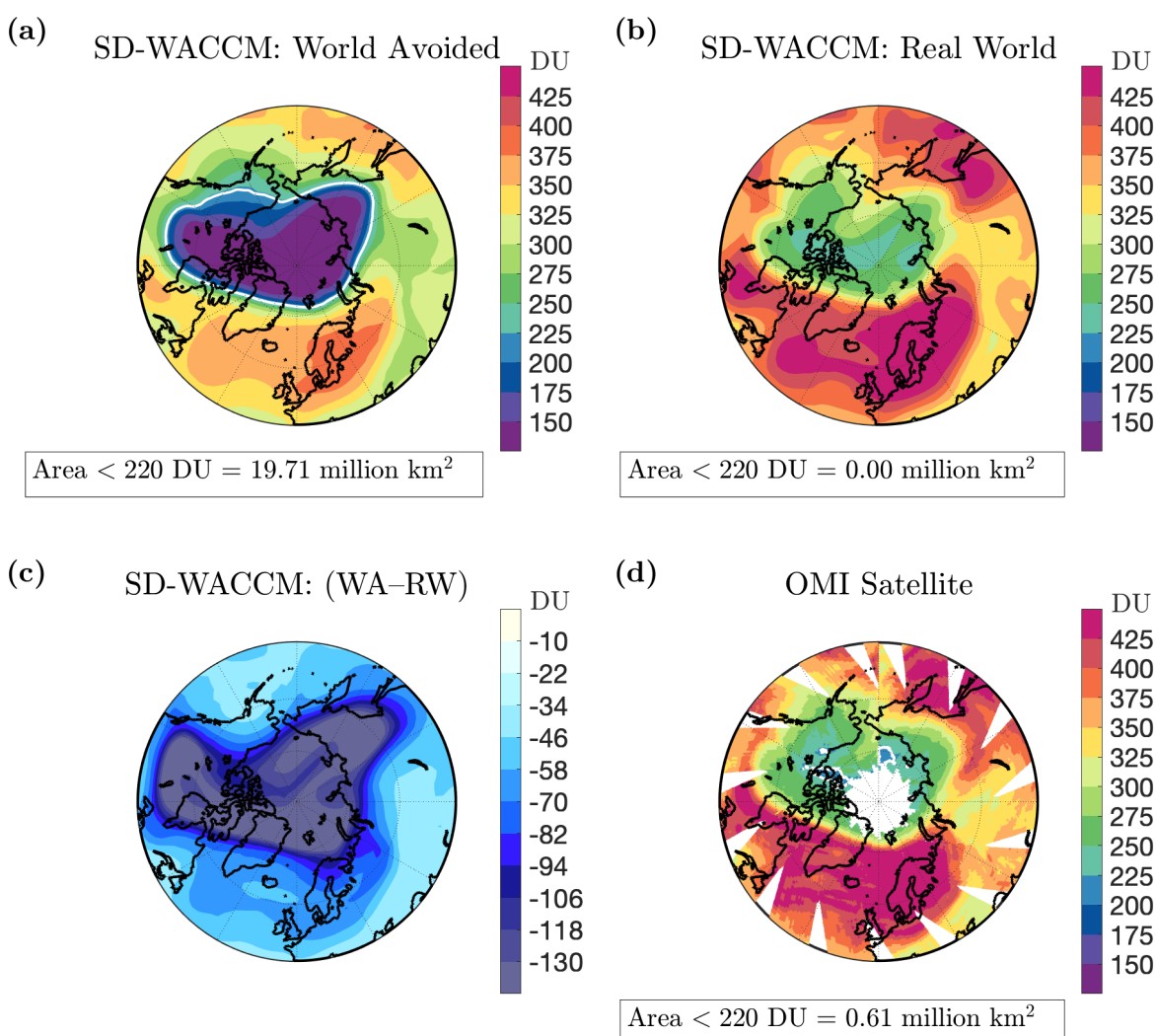

**Figure 1.** Total Column Ozone poleward of 30° N for March 13, 2020. The upper left figure shows the World Avoided SD-WACCM run, the upper right figure shows the Real World run, and the lower left shows the difference between them. The lower right figure shows the Total Column Ozone Level 3 product from the OMI satellite. All levels are in Dobson Units. Note the different scale on the lower left colorbar. The 220 DU contour is outlined in white.

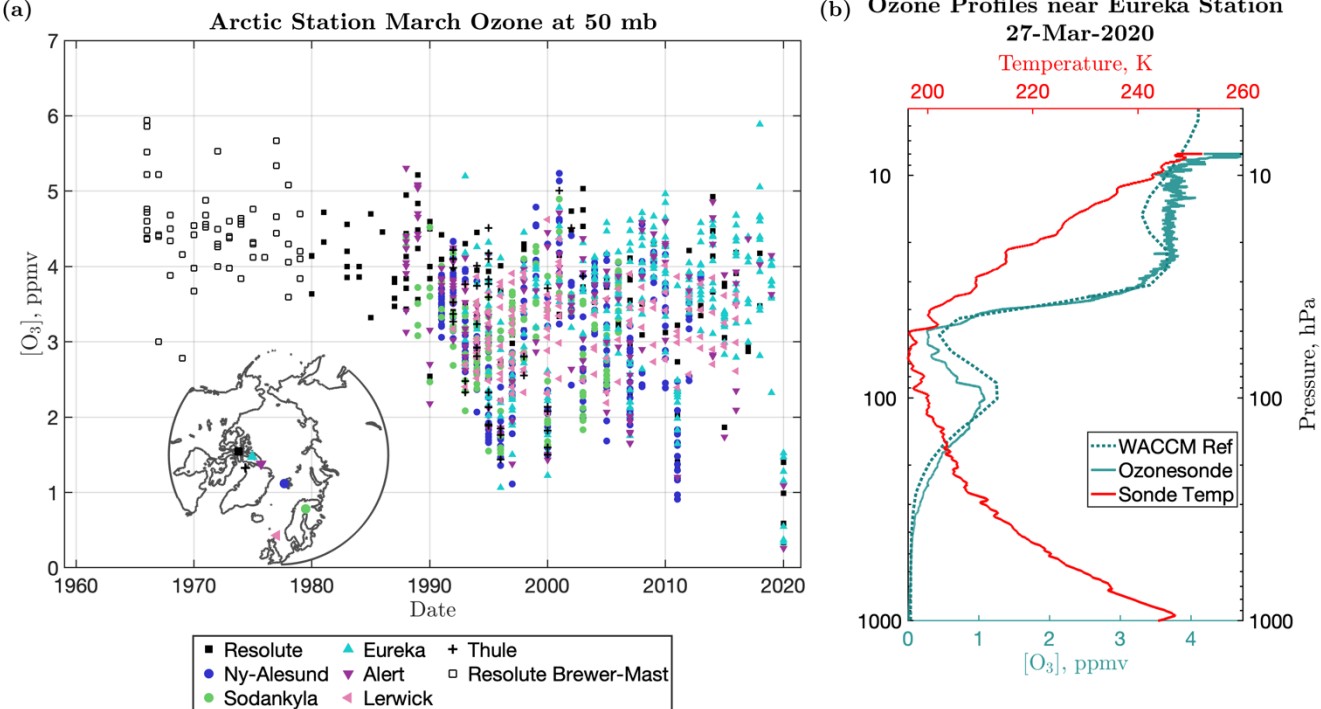

**Figure 2. (a)** Daily ozone values centered at 50 mb (±2.5 mb) from ozonesondes launched from various stations across the northern polar region in March. Measurements using less accurate methods are indicated with open symbols. The location of these stations is shown in the lower left corner of the panel. **(b)** Ozone (solid teal) and temperature (solid red) profiles taken at Eureka station on March 27[th], 2020, compared to the SD-WACCM Real World run's (dotted teal) vertical profile at the nearest model gridpoint.

The right panel of Figure 2 compares the vertical profile of ozone at the nearest SD-WACCM grid point in the RW simulation (dotted teal line) to Eureka station ozonesonde data (solid teal line) for March 27[th], 2020, showing some of the lowest values of stratospheric ozone ever recorded in the Arctic. Eureka is near the center of the lowest total ozone on this date and is representative of the region based upon the model, and on comparisons with other high-Arctic sites, which show similar profiles. The figure shows that the largest depletion here tracks the lowest local temperatures of the profile (temperature shown in solid red). Although temperature histories can also be important, as activation can persist in air parcels which previously encountered cold air but are currently above the temperature threshold for PSC formation, this broadly supports the view that much of this year's ozone loss was related to widespread local cold temperatures increasing

the efficiency of heterogeneous reactions on PSCs (Wohltmann et al., 2020; Manney et al., 2020). Figure 2 illustrates that the SD-WACCM model successfully captures the observed behavior at this site under these extreme conditions.

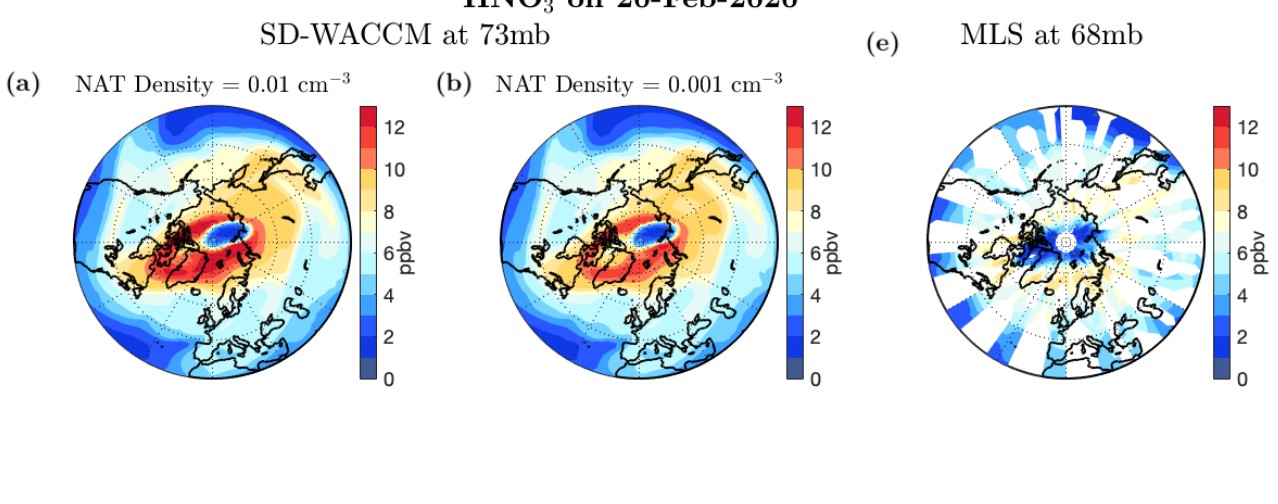

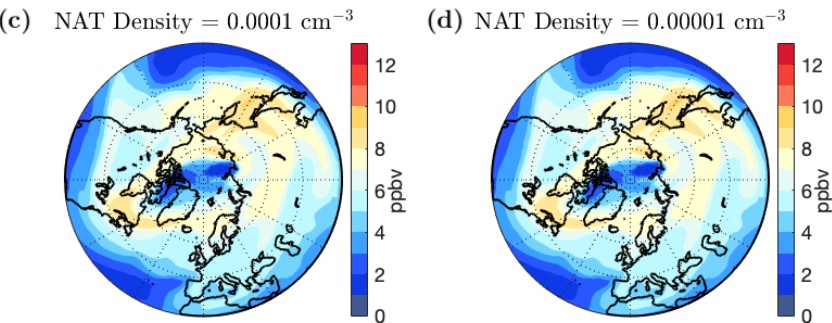

Figure 3. HNO₃ in ppbv for February 20th, 2020, for different NAT parameterizations in SD-WACCM (a-d) compared to MLS (e). In order of increasing denitrification, the NAT density is (a) 0.01 cm⁻³, (b) 0.001 cm⁻³, (c) 0.0001 cm⁻³, and (d) 0.00001 cm⁻³. Panel (a) shows the previous standard SD-WACCM parameterization and panel (d) shows the chosen parameter value used in the RW and WA simulations. All SD-WACCM figures show the 73 mb level; MLS shows the 68 mb level.

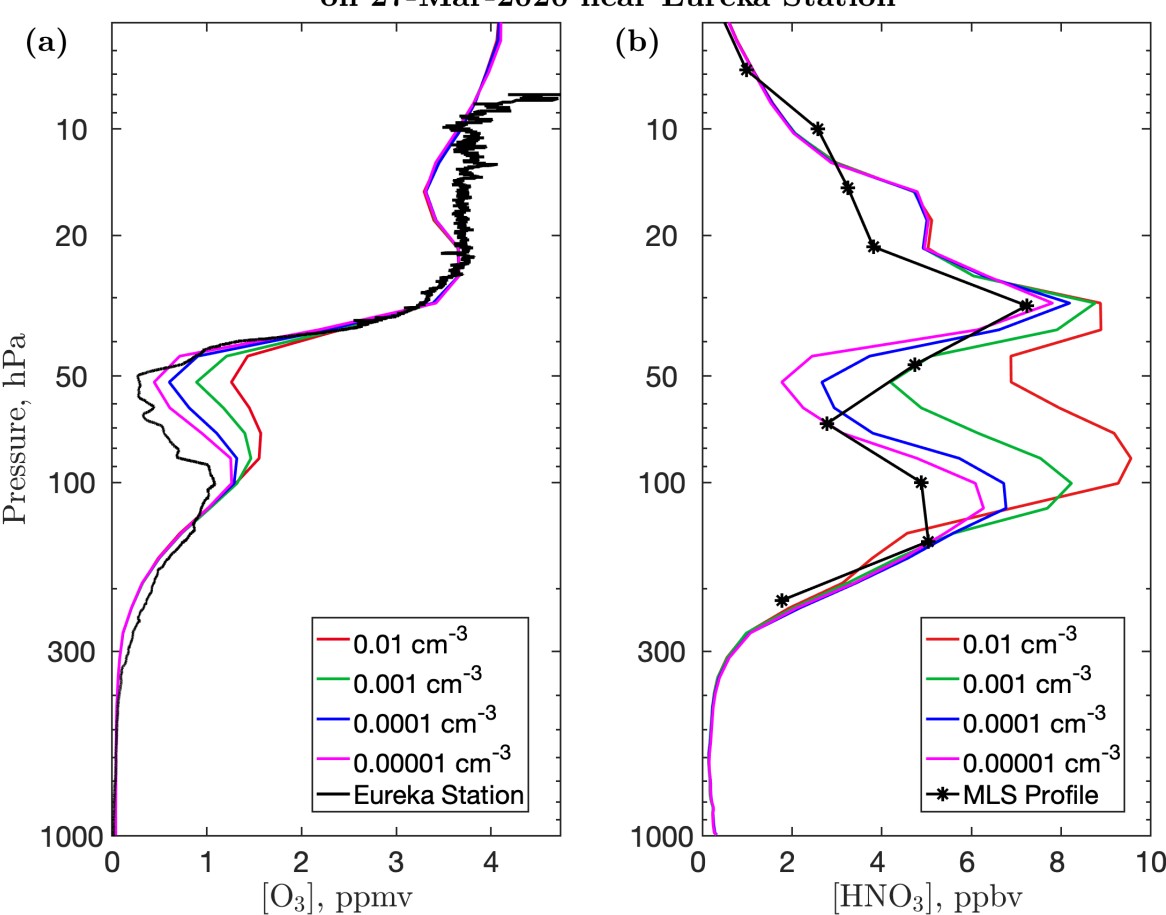

**Figure 4.** Comparison of ozone profiles **(a)** and nitric acid profiles **(b)** for four different SD-WACCM simulations at the gridpoint nearest Eureka Station for March 27[th], 2020, with ozonesonde data from Eureka Station shown for comparison in **(a)** and the nearest MLS profile on that day shown for comparison in **(b)**. Note that MLS has very few levels in the lower stratosphere, shown by black points. This is the same date as shown in Figure 2b, and the magenta ozone profile corresponds to the dotted teal ozone profile shown in that figure.

We next test how SD-WACCM's nitric acid trihydrate (NAT) number density count relates to calculated denitrification (Fahey et al., 2001) using comparisons to nitric acid observations from the Microwave Limb Sounder (MLS) instrument onboard the AURA satellite (Waters et al., 2006) and ozonesonde profiles. Polar stratospheric clouds not only activate chlorine through heterogeneous chemical processing but also denitrify the atmosphere through removal of $HNO_3$ from the gas phase and subsequent sedimentation. Removing $HNO_3$ reduces the abundance of $NO_2$, which in turn enhances active

chlorine (i.e., ClO abundances) by reducing $ClONO_2$ formation rates, affecting ozone destruction chemistry. Initial comparisons of ozone profiles to both ozonesonde and MLS data showed that the model's standard approach with a NAT density of 0.01 $cm^{-3}$ (Wegner et al., 2013) was denitrifying too little. As a lower NAT particle density corresponds to larger individual particles, decreasing this parameter increases denitrification by increasing the settling velocity of the particles.

Figure 3 shows the progression of four increasingly denitrified RW runs from (a) to (d).

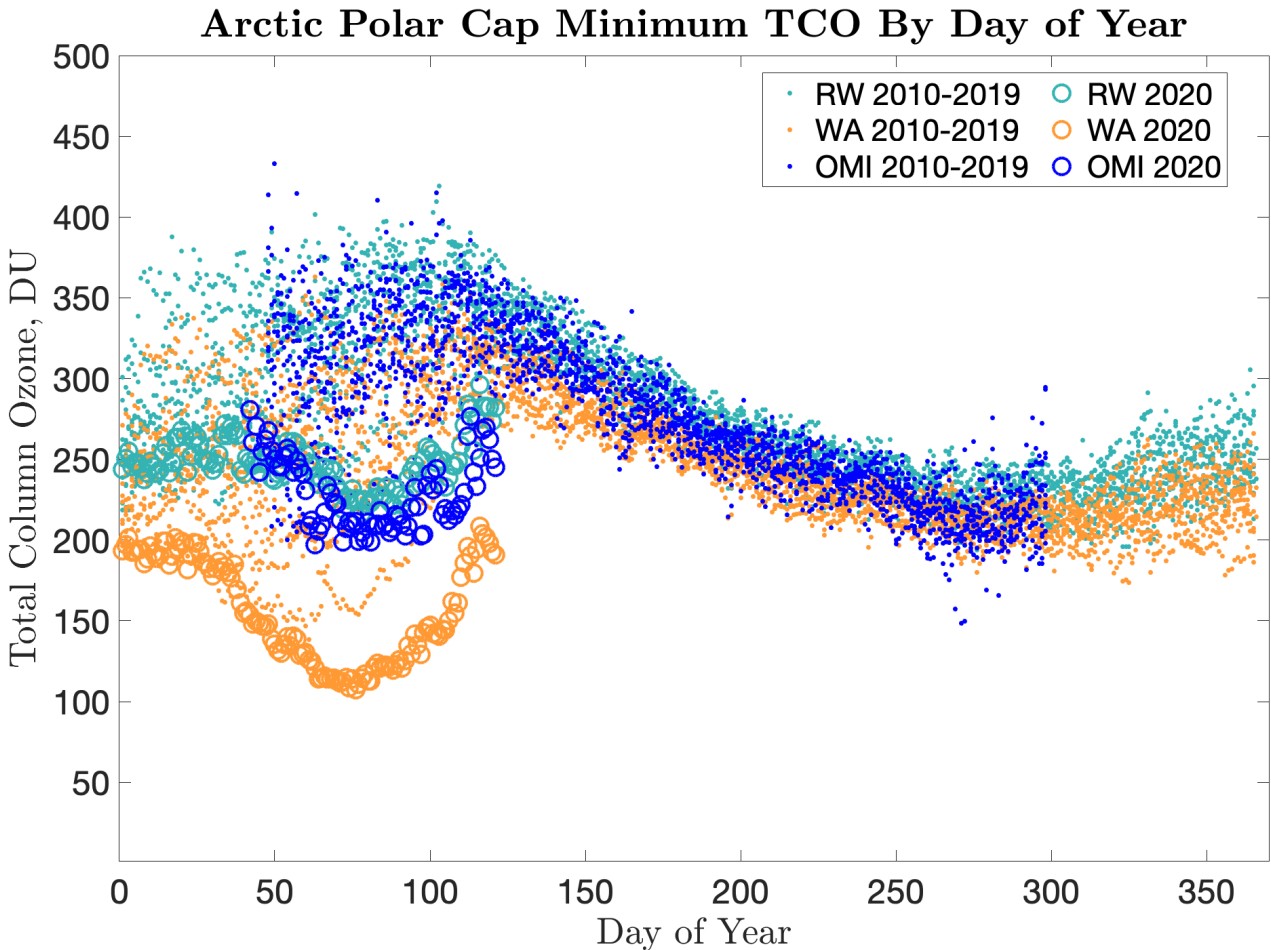

**Figure 5.** Minimum total column ozone simulated by SD-WACCM from 70° N to 90° N from January 2010 through the end of April 2020, plotted by day of the year. Teal markers refer to the reference run and orange markers to the world avoided run. Blue markers refer to observations by the OMI Satellite. Dots indicate days from 2010-2019 and open circles indicate

215    days in 2020.

Although the coverage of MLS data swaths (e) makes it difficult to distinguish which of the two highest ((c) and (d)) denitrification levels might be a better representation, the two lower denitrification levels (panels (a) and (b)) are much

poorer matches to the observations. We ultimately found that a case adopting a NAT particle density of $10^{-5}$ particles per $cm^3$ resulted in the closest match to observed ozone profiles throughout a wide vertical range throughout the spring (Figure 4a for

Eureka on March 27, and Figure S4 for other times at Eureka along with the Alert and Resolute stations) and matched the observed MLS $HNO_3$ profiles better than other choices (Figure 4b), and so chose this value for our RW and WA simulations. As all of our RW runs adopt observed temperatures insofar as they are represented by the MERRA2 reanalysis, this study illustrates that accurate representation of denitrification (i.e., not only accurate temperature-driven reaction efficiencies but also PSC microphysics impacts) is key for ozone depletion under 2020 Arctic conditions.

In Figure 5 we examine the evolution of daily minimum TCO by day of the year from January 2010 to the end of April 2020 for the polar cap north of 70°N, with days in 2020 marked as open circles rather than points. We compare the RW (teal markers) to observations from the OMI satellite (blue markers) and compare both with the WA (orange markers). Dramatic differences are obtained in the calculated and observed 2020 evolution of the daily minimum TCO value over the Arctic polar cap by day of the year for the past decade in Figure 5 compared to the preceding years and especially for the World

Avoided. Prior to 2020, while the WA case is often lower than the other two, it is still within the range of TCO values seen in the RW and OMI time series. Furthermore, both the RW run and the OMI observations for 2020 spring display lower values than many WA springs, illustrating the key role of the unusually cold temperatures in addition to chlorine in driving the depletion in 2020. The WA spring of 2020 both displays levels of depletion previously unseen in the data or either simulation early in the spring and stays depleted longer than any other year. Furthermore, its apparent dip compared to the

rest of the year resembles typical Antarctic ozone evolution (shown in Figure S5) rather than the typical Arctic behavior. The effects of higher chlorine loading in the WA scenario on vertical ozone profiles are also significant (Figure 6, left panel). While both the RW run and ozonesonde data display a limited height region of extremely low ozone, the World Avoided has almost no ozone left throughout the lower stratosphere. This resembles typical Antarctic depletion more than any previous year in the Arctic (Figure 6, right panel, with ozonesonde comparison). Depletion in the lower stratosphere reaches these low

values more quickly in the WA and persists longer (Figure S6). At higher altitudes, where gas phase depletion identified by Molina and Rowland (1974) is dominant, substantial increases in depletion are also obtained (see below).

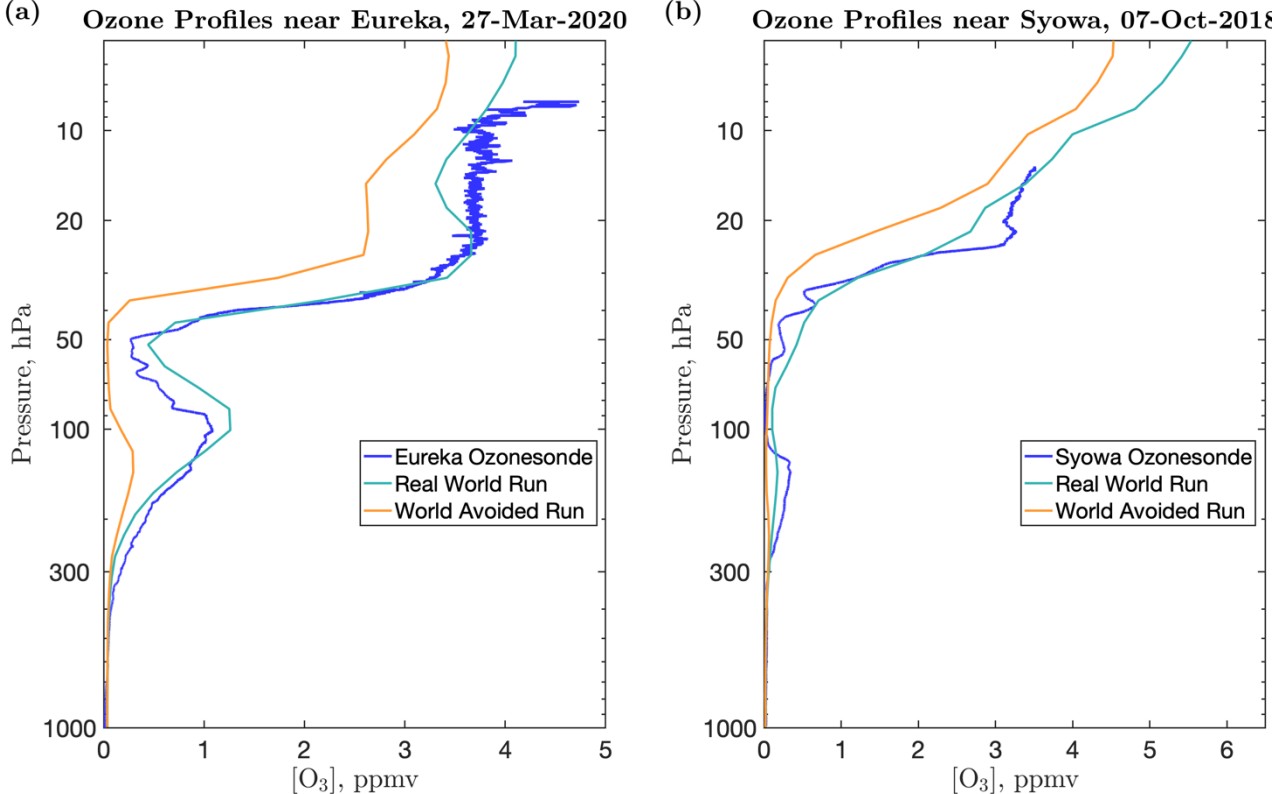

**Figure 6.** Comparison between the Real World (teal) and World Avoided (orange) ozone profiles in SD-WACCM at the gridpoint **(a)** nearest to Eureka station (80.04°N, -86.18°E) for March 27[th], 2020 and **(b)** nearest to Syowa station in the Antarctic (69.00°S, 39.58°E) for October 7th, 2018. Ozonesonde profiles from the stations are shown for comparison in blue.

A characteristic finding of WA studies is that substantial polar ozone depletion eventually persists year-round (Newman et al., 2009; Garcia et al., 2012). We can see the first indication of such behavior in our WA simulation for 2020 as shown in Figure 7, where the RW and WA total ozone time series are shown for the past decade (top panel), along with their difference (bottom panel). While for the first few years of the decade the summertime and autumnal differences between the scenarios remain low and fairly constant, after 2014 a noticeable trend towards increasing column difference year-round emerges. Much of this summertime difference is due to the gas-phase depletion, as demonstrated by the change in the profiles and increased partial column differences at higher altitudes (Figure S7). It is also noteworthy that while the spring of 2020 is anomalously depleted in the WA as previously shown, the spring ozone values obtained in 2018 and 2019 are also much further from their RW counterparts, demonstrating the growing impact of the Montreal Protocol even for less cold years.

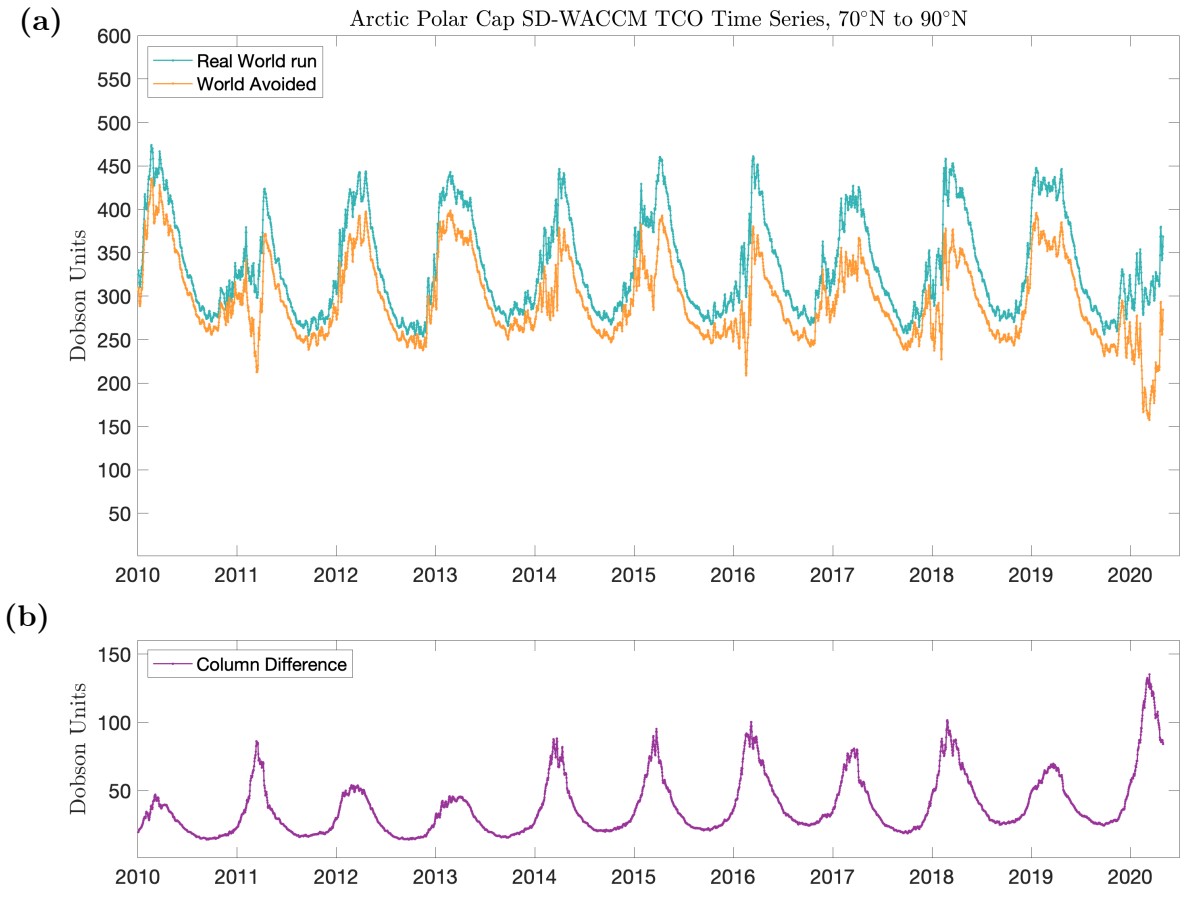

**(a)**

Arctic Polar Cap SD-WACCM TCO Time Series, 70°N to 90°N

**(b)**

**Figure 7. (a)** Time series of mean SD-WACCM total column ozone across the polar cap for the Real World scenario (teal) and World Avoided scenario (orange) from January 2010 through April 2020. **(b)** The difference between the two.

## 4 Discussion and Conclusions

We have demonstrated that, were it not for the Montreal Protocol, the meteorological conditions seen in 2020 would have produced the first Antarctic-like ozone hole over the Arctic, an area with a substantial human population and vibrant ecosystem." The Arctic ozone hole would have begun earlier and persisted longer (see Figure S6) than the headline-grabbing 2020 ozone depletion in the real world did, with ozone all but completely destroyed over a large vertical extent of the lower stratosphere. Furthermore, our simulations support the view that there have already been substantial year-round benefits from the Montreal Protocol for the Arctic. Finally, nitric acid observations and modelling for 2020 help improve our

understanding of the role of denitrification in accurately assessing Arctic ozone loss, and further refinements of this will be the subject of future studies.

The main limitation of using a model constrained to real-world meteorology is that it by design eliminates any feedbacks that changes in ozone would have had on the meteorology. These are worthy of investigation in future free-running model simulations, especially in the context of potential increasing stratospheric Arctic cold extremes from climate change, which

have been debated in the literature (Rex et al., 2004). In addition to the increased radiative forcing from increasing ODSs, ozone itself is a potent local greenhouse gas, and ozone depletion of the magnitude simulated here would significantly alter the temperature profile in the stratosphere and perhaps in the troposphere as well. Resultant changes in stratospheric dynamics could potentially have then led to changes in surface climate and sea ice (Smith et al., 2018; Stone et al., 2019; Stone et al., 2020). Surface UV increases could be especially important during the Arctic summer, when the vast majority of

biological growth takes place. A coupled biosphere model would be required to fully investigate such effects, but we can estimate the change in surface UV during late March and April 2020 by calculating the clear-sky UV Index at noon for the SD-WACCM grid-point nearest four northern cities using the procedure outlined in Burrows et al. (1994), with the results shown in Table 1. While the UV Indices at the end of March in the WA run show substantial percentage increases (from 25% to 88%), the absolute values are still quite small due to the large zenith angles in the Arctic spring. By the end of April,

however, there are still substantial differences between the runs, despite active ozone depletion having ceased for the year, reinforcing that there are increasing year-round impacts, as seen in Figure 7.

|  | RW MARCH 31 | WA MARCH 31 | RW APRIL 30 | WA APRIL 30 |
|---|---|---|---|---|
| **FAIRBANKS** | 1.15 | 1.44 | 2.49 | 3.06 |
| **YELLOWKNIFE** | 1.62 | 2.52 | 3.05 | 3.75 |
| **TROMSØ** | 1.12 | 1.93 | 2.32 | 2.75 |
| **MURMANSK** | 1.27 | 2.39 | 2.19 | 2.66 |

**Table 1**. Surface UV Index for the SD-WACCM grid-point nearest to Fairbanks, USA (64.84°N, 147.7°W), Yellowknife, Canada (62.46°N, 114.22°W), Tromsø, Norway (69.66°N, 18.94°E), and Murmansk, Russia (68.96°N, 33.08°E). UV Index

is calculated for March 31 and April 30 for both the RW and WA simulations using total column ozone and solar zenith angle at noon under clear-sky conditions in each grid-point.

The benefits to society and the earth system achieved by the global community's adherence to the Montreal Protocol grow with each passing year and can be dramatically documented in cold years with ozone depletion-favoring meteorology – in particular, 2020. As we progress further into the 21st century, studies of the world we avoided will continue to be relevant to

295 both stratospheric science and environmental policy. When the Montreal Protocol was signed, the sophisticated modeling

systems used for this and similar studies that can precisely simulate an alternate world did not yet exist. The basic science was, however, sound enough, and the risk clear enough, that society acted nonetheless. Here we have shown that our increased knowledge of what we would have faced has justified this past prudence.

**Author Contributions:** D.K., S.S., and C.W. contributed to the modelling run setup and interpretation. D.T. contributed to data interpretation. All authors contributed to writing the paper.

**Competing Interests:** The authors declare that they have no conflict of interest.

**Acknowledgments and Data Availability:** C. W. and S. S. were partly supported by a gift from an anonymous donor to MIT. D. K. was funded in part by NASA grant (80NSSC19K0952). WACCM is a component of the CESM, supported by
305 the National Science Foundation (NSF). We would like to acknowledge high-performance computing support from Cheyenne (doi:10.5065/D6RX99HX) provided by NCAR's Computational and Information Systems Laboratory, sponsored by the NSF. MERRA2, OMI, and MLS data can all be freely obtained online through NASA (https://gmao.gsfc.nasa.gov/reanalysis/MERRA-2/data_access; https://disc.gsfc.nasa.gov/datasets/OMTO3e_003/summary; https://mls.jpl.nasa.gov/products/hcl_product.php). Ozonezonde station data can be accessed through the World Ozone and
310 Ultraviolet Data Center (WOUDC, https://woudc.org/data/explore.php). Model results shown in this paper are available online (at: https://acomstaff.acom.ucar.edu/dkin/ACP_Wilka_2020/).

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
