# Peer review of "An Arctic Ozone Hole in 2020 If Not For the Montreal Protocol"

_Atmospheric Chemistry and Physics, 2020_

## Author Comment (AC1)

We would like to thank the two referees for their helpful feedback and comments. Our changes to both text and figures in response to their suggestions are summarized below. We have also responded to each Referee Comment point in detail in green text below.

Figure Changes:

1) We have increased the text size and linewidth in Figures 1, 2, 3, 5, and 6, as well as in Supplemental Figures S1, S2, S3, and S5.

2) We have changed the color-scale for panel (c) in Figures 1 and S2, as we do not want red values to be inadvertently seen as enhancement rather than greater depletion.

3) We have added Figure 4 to show the impact of changes to the NAT parameterization on ozone and HNO3 profiles directly.

4) We have added Figure S4 to show additional ozone profile comparisons for other days and stations in Spring 2020.

5) We have added ozonesonde data to Figure 6 (previously Figure 5) to better illustrate the differences between typical Arctic and Antarctic depletion profiles.

Major Textual Changes:

1) We have clarified our explanation of the physical importance of the nitric acid trihydrate (NAT) parameterization changes in the Abstract, Introduction, and Methods section. We have expanded our discussion of Figure 3 and the new Figure 4 to better justify our chosen modifications.

2) We have calculated the expected changes to surface clear-sky UV for several prominent northern cities for March 31, 2020 and April 30, 2020. We present these changes in Table 1 and have expanded the Discussion and Conclusions section to include more consideration of these effects.

**Referee #1 Comments and Response**

There is a great deal of interest in the extent of ozone depletion during the recent cold Arctic winter of 2020, with a number of papers appearing, especially in a JGR/GRL special issue. This paper adds to that work mainly by investigating how much worse the depletion would have been without the controls of the Montreal Protocol. This is done through a specified-dynamics 3-D model run with a scenario that assumes no protocol. Secondary aspects of the paper are presentation of Arctic ozone sonde observations (including 2020) and an investigation of different model denitrification schemes.

I find that that paper has some interesting results would could make it suitable for publication in ACP. However, I do find that further analysis is needed in order to back-up some of the conclusions, along with clearer organization of the main points. I give my comments below.

General Comments

1) One main message of the paper is that the Arctic ozone depletion of 2020 would have been much worse without the Montreal Protocol. That is without doubt. However, it is not possible to know exactly how demand for CFCs and similar gases would have evolved. The 3.5%/year growth since 1985 is an assumption and that should be made clear.

We agree that the exact level of ODS growth chosen is only one hypothetical pathway. We now make this fact clear and explain how we arrived at the 3.5% value in Lines 75-78:

"The assumption of 3.5% per year growth matches that used in the Garcia et al. (2012) World Avoided study and is a good approximation of the grown rates seen in years immediately prior to emissions controls, thus representing an illustrative "business as usual" alternate trajectory."

2) It seems like more use can be made of the available sonde data for evaluation of the Real World run. There are ozone plots with model only (5, S6a), and plots with data only (2a, S3). Better use of the data could be made for evaluating the model.

We agree that more usage of sonde and other observational data strengthens the paper. We have added ozonesonde profiles to Figure 5, and now include a new figure (Figure 4) showing the direct comparison of different denitrification parameterizations to both ozonesonde data and MLS $HNO_3$ profile data, which is described further in our response to General Comment (3). We have also added a new supplementary figure (Figure S4) showing the comparison of the model to sonde data taken at Alert, Eureka, and Resolute for three days in the spring each to provide a more comprehensive review of how the new parameterization performs. The explanation of these new plots is presented in the Results section, Lines 207-225.

3) One message from the abstract is that the large Arctic ozone depletion of 2020 can be used to test the parameterization of PSC denitrification. Here the results presented do not go into enough detail. Maps of HNO3 are shown for 70 hPa. A number of questions come to mind. How well do the simulations do at other altitudes? How would this affect other winters? I realise that this are major questions but normally a study which aims to present an improved denitrification model would be based on more than just one altitude in one winter. Also, the ultimate choice of the denitrification scheme is based on the impact on ozone which is not shown. This is an indirect test and we need to see how large the sensitivity of ozone is.

We agree with the Reviewer that further explanation is needed to justify our choice of denitrification scheme. To that end, we have added in a figure (Figure 4) showing the comparison of ozone (4a) and nitric acid (4b) profiles that alerted us to the problem with our denitrification approach and led us to our final choice of scheme. We have included ozonesonde and MLS profile data as comparisons and believe that this new figure shows the improvement in both ozone and nitric acid over multiple levels in the lower stratosphere. We have also added a supplementary figure (Figure S4) showing the equivalent of Figure 4a for three days each at the stations Alert, Eureka, and Resolute, which we believe provides more confidence that the new parameterization is an improvement. (While we tried to space the comparison days out evenly, we ultimately prioritized days for which sondes reached above 50mb, so not all of the stations show data from the same days.) We have updated the text to reflect these change and expanded our justification in the Results section. Finally, we have clarified which NAT parameterization was originally in use and which one we ultimately chose in both the Methods (Lines 99-101), and Results (Lines 208-225) sections, as well as in the caption of Figure 3.

4) It is not clear if the paper is presenting the 2020 Arctic sonde data for the first time. Is that a novel aspect of this paper?

Wohltmann et al., 2020 was (to our knowledge) the first to present a systematic comparison of Arctic 2020 sonde data from multiple stations, and several studies have since focused on individual locations. We have updated the end of the Introduction to clarify this, and it now reads on Lines 66-69:

"The Arctic spring of 2020 displayed very cold temperatures and a stable polar vortex that led to record levels of Arctic PSCs, and deep Arctic ozone depletion in the real world at some altitudes in the lower stratosphere, as has already been shown using both ozonesondes (Wohltmann et al., 2020) and satellites (Manney et al., 2020)."

Specific Comments

5) I think that the abstract needs a lot of work. It is short compared to what is possible in ACP and it lacks some details. Also, it seems to jump around in the topics covered. The abstract mentions 'record observed local lows'. It is not said where these numbers are from and if this paper is presenting the 2020 sonde data for the first time. The abstract also says that 'This provides an opportunity to test…' without stating how the parameterizations are being tested and what the results are. It is also not clear that it is the RW run which is used for the testing; the abstract makes it sound like the WA run allows the testing. After this, the abstract returns to the WA run so the summary of that is split. Overall, I think that the abstract needs a more logical flow to cover the results and more information to summarise what was found.

We have reformatted the abstract by adding more details about the model and instrument comparisons, expanding the summary of our parameterization changes, and grouping more of the WA results together. We agree that this improves the logic and flow of the abstract and gives the reader a better summary of what we present in our paper.

6) Line 23. Farman et al.

This has been corrected.

7) Line 24. 'PSCs were the primary culprit'. This is not really true. PSCs are an essential part of the chain of events which contains a few steps. CFCs (and other chlorine gases) could be seen as a culprit and the

one that humans can control. Alternatively, PSCs could be described as the key step that was not understood and why the ozone hole was not predicted etc.

We have rephrased this and linked PSCs more explicitly to denitrification as follows in Lines 27-32, which we agree improves the logical flow of this explanation:

"Subsequent work (Solomon et al., 1986) revealed that heterogeneous chemical reactions involving chlorine and bromine on the cold surfaces of Polar Stratospheric Clouds (PSCs) were the missing link in the sequence of steps leading to this deep depletion. PSCs are made of water ice, nitric acid trihydrate (NAT), or supercooled ternary solutions of water, nitric acid, and sulfur (STS), and several studies have highlighted a significant role for sedimentation of large NAT particles in the removal of HNO3 from the lower stratosphere, or denitrification (Toon et al., 1986; Crutzen and Arnold, 1986)."

8) Line 39-40. 'near-complete recovery'. What does this mean? We don't expect a smooth return to e.g 1980 ozone levels everywhere. Models suggest the tropics may not get to those levels before column ozone starts to decrease due to circulation changes.

The WMO 2018 Scientific Assessment of Ozone Depletion predicts a worldwide return to near 1980s values for ozone as a result of the Montreal Protocol specifically limiting ODSs. We recognize that there are other anthropogenic factors which could alter the background state in a different manner, but as these are highly dependent upon selected emissions trajectories and uncertain future policy decisions, we are restricting our analysis to the success of the Montreal Protocol itself. We have added a clarification to our statement of near-complete recovery, as well as the WMO 2018 reference, to Lines 48-50 that we believe more explicitly places it within this context:

"the world appears to be on track for near-complete ozone recovery to near 1980s values as a result of decreasing ODSs by the second half of the 21$^{st}$ century (WMO 2018), and the Protocol has been ratified by every state represented at the United Nations"

9) Lines 68-69. You should make it clear that this scenario of 3.5% year growth since 1985 is an assumption and state why you choose these parameters.

We have clarified and expanded upon this choice as explained in our response to General Comment 1, with an emphasis on the fact that this is a scenario assumption only.

10) Line 87. Why do you use the smallest value for the WA runs? Please explain.

We agree that adding physical justification for our NAT parameter choice is necessary, and now include a sentence to that effect in Lines 204-205:

"As a lower NAT particle density corresponds to larger individual particles, decreasing this parameter increases denitrification by increasing the settling velocity of the particles."

Additionally, we have clarified that we chose the smallest value for both the final RW and WA runs based upon our testing of multiple NAT parameter choices. We now better differentiate between the prior value in SD-WACCM and our chosen value in both the Methods and Results sections, as explained in the response to General Comment 2.

11) Line 104. Waters et al seems to be a general MLS reference. Please also give one for the specific HNO3 product.

We have added a reference to Lambert et al., 2007, which is an HNO3-specific validation study.

12) Line 112. Spell out WOUDC.

This change has been made.

13) Lines 113. Give the location (latitudes at least) of all the stations here.

We have added locations of all stations as (latitude, longitude) in the text, and added the coordinates of the stations used in Figure 6 in the caption of that figure.

14) Lines 118-119. The paragraph ends on a confusing not because it is stated that Eureka is used for profile comparisons, but that leaves the reader wondering what the other sonde data is used for (time series at 50 mb, as it turns out).

We now specify in Line 129 that other station data is used for the time series at 50 mb so that the reader has this information earlier.

15) Lines 133-134. Here figure for the size of the 2011 Arctic ozone 'hole' is given – 11 million km2. This is still significant (see statement on what is an ozone hole on line 59) but I think that the authors point is that it is smaller than that which has occurred in the Antarctic since the mid 1980s when the term 'ozone hole' was first used? These points need to be made clear here in the results section. This also relates to the title of the paper, which presents 2020 as the first time that historic meteorology would have led to an 'ozone hole'. A big factor in this change since 2011 is clearly the assumed increasing chlorine, but also the meteorology of 2020 had, I believe, anomalously low dynamical ozone replenishment. Can you comment on the importance of these factors? There are at least references for the low absolute column ozone in 2020 in the JGR/GRL special issues.

We now comment on the dynamical differences between 2011 and 2020, specifically the fact that 2020's polar vortex lasted longer than in 2011, with new references to two recent studies that compared these in great detail, as shown in Lines 147-150:

"The difference is partly due to the increased chlorine loading in the WA, but we also note that, while 2011 was an extremely cold year, 2020 had lower minimum ozone values that lasted longer than in 2011, with record fractions of the polar vortex being below the PSC temperature threshold for a longer period of time (Wohltmann et al., 2020; Inness et al., 2020)."

16) Line 147. Figure 1. Please use (a), (b) etc here to label the panels.

Figures 1 and S2 have been updated with letter labels for the panels and the subpanels have been updated with larger text for clarity.

17) Line 157. Caption for (b) should make it clear only model O3 is plotted.

We are plotting both the model ozone profile (dotted teal line) and the Eureka station ozonesonde ozone profile (solid teal line), along with the ozonesonde temperature profile (red line). We now reference each line explicitly in both the caption and the text to avoid confusion.

18) Line 158-159. Confusing as worded. The sonde, which shows the record low O3 values, is at Eureka, and then compared to the nearest model profile.

We have reworded this figure description and now state which line color corresponds to which profile, which we believe improves the readability of the figure.

19) Line 167. Better to say 'instrument' not 'satellite'.

This has been corrected.

20) Line 170. The figure caption is not very specific to the individual panels. Better to call out each individual panel and state what is changing in each case.

We have changed the figure caption to include this information and also refer to each panel in turn in the text.

21) Line 174-175. Please indicate which of the runs shown is the 'standard' WACCM approach.

The standard WACCM approach, with a NAT parameter value of 0.01, is now labeled as such in both the text (line 209) and the caption of Figure 3.

22) Line 175. Six? Figure 3 has 4 model panels? Also, could use labels (a)-(d) rather that saying 'first two rows'. That seems tidier.

We agree that Figure 3 benefits from easier-to-read labels and have redone each panel. The references to model panels in Figure 3 have also been clarified and corrected, with each of the four model panels and the fifth MLS panel now referred to by (a)-(e) in the text.

23) Line 177. Figure 4 caption. This continues the pattern of the figure captions being too brief and not fully explaining the plot. The captions starts by calling out the 2020 points but these only extend from January to April. Also, the points 'prior to 2020' are, more specifically, years 2010-2019 (based on title). Therefore, I would suggest first describing the majority of the plot (minimum ozone by day of year for 2010-2019), and then describe the 2020 points. The left hand column in the legend should have the dates (2010-2019) to make clear the distinction with the right-hand column.

We have reworded this caption to make what is being shown clearer. Specifically, we have corrected the dates, and now refer to all years with the term "markers" rather than "points," which we believe was needlessly confusing, given that some of the markers have the form of dots/points. We now explicitly differentiate what the open circles vs dots show and have changed the legend as suggested. We have also redone this panel with larger markers, which we believe improves the visual distinction of the different time series.

24) Line 180. 'bottom row, middle panel'. There is only 1 panel? I would suggest using '(e)' in any case.

This has been corrected, as detailed above in response to (20) and (22).

25) Line 181. three?

This has been corrected, as detailed above in response to (20) and (22).

26) Line 182-183. There is a major leap in the argument here. Figure 3 is comparing model v MLS HNO3 to test the denitrification, which impacts HNO3 directly. Then, the argument jumps to using the O3 profile to test the model. There are many other processes which would affect the model ozone profile (chemistry, dynamics) and so there needs to be much better justification for this.

As we describe in our response to General Comment 3, we agree and have added new plots (Figures 4 and S4) and expanded the denitrification discussion to address this point.

27) Line 184. 'insofar as they can be determined from reanalysis'. Better to say 'insofar as they are represented by the reanalyses'.

This has been corrected.

28) Line 188 'dailyminimum' – space needed.

This has been corrected.

29) Line 188-190. This is a long sentence and difficult to read. It is not clear when it mentions 'for the past decade … compared to the preceding years'. Please revise.

We agree that this sentence needed clarification. The description of Figure 5 (previously Figure 4) has been reworked to present the important information more clearly, as detailed further in the response to Comment 30 below.

30) Line 190-191. 'the scatter of the three overlaps'. The RW run and OMI should ideally have the same scatter! Better to compare RW versus OMI, and if the model is good you can compare RW v WA.

We have expanded the description of Figure 5 (previously Figure 4) in Lines 228-230 and separated out the descriptions of the different time series more, which we believe results in a more readable paragraph:

"In Figure 5 we examine the evolution of daily minimum TCO by day of the year from January 2010 to the end of April 2020 for the polar cap north of 70°N, with days in 2020 marked as open circles rather than points. We compare the RW (teal markers) to observations from the OMI satellite (blue markers) and compare both with the WA (orange markers)."

Furthermore, we have rephrased the discussion about scatter, as it was initially meant to refer to the fact that pre-2020 WA points often were within the typical range of TCO seen in the RW and OMI measurements, with the additional chlorine not driving significant additional depletion. We believe the reworked text (Lines 226-230) now makes this clearer:

"Prior to 2020, while the WA case is often lower than the other two, it is still within the range of TCO values seen in the RW and OMI time series. Furthermore, both the RW run and the OMI observations for 2020 spring display lower values than many WA springs, illustrating the key role of the unusually cold temperatures in addition to chlorine in driving the depletion in 2020. The WA spring of 2020 both

displays levels of depletion previously unseen in the data or either simulation early in the spring and stays depleted longer than any other year."

31) Line 194. Figure 5 caption. The bold/non-bold text is reversed compared to other figures. The caption should state 'ozone' somewhere. Give the latitudes of the stations so that it is clear Eureka is north and Syowa is south. These are sonde stations so please show the data for comparison (or choose a close day when there is data).

We have fixed the caption text formatting and added ozonesonde data from the stations themselves to these plots, along with the locations of the station. In doing so, we had to change the day shown for the Antarctic from October 1, 2018 to October 7, 2018 due to data availability.

32) Line 196. I don't think you can use 'record' for a hypothetical model run? A larger assumed use of CFCs would give even lower O3!

We have changed the phrasing for this sentence to clarify the context in which the WA depletion takes place and what we are comparing against in Lines 229-230:

"The WA spring of 2020 both displays levels of depletion previously unseen in the data or either simulation early in the spring and stays depleted longer than any other year."

33) Figure S1. Please give a brief explanation of the 'total equivalent effective chlorine'. I assume this is the tropospheric equivalent chlorine loading? State the alpha factor used for the calculation of equivalent chlorine. What does 'effective' mean in this context?

We are showing the stratospheric equivalent chlorine loading, which is calculated as a linear combination of $Cl_y$ and $Br_y$, with $Br_y$ scaled by a factor of 60 in the midlatitudes and 65 in the polar regions. We have updated the figure caption of Figure S1 with this explanation.

**Referee #2 Comments and Responses**

The paper by Wilka et al. examines the significant Arctic ozone depletion that occurred during spring 2020 using the Specified Dynamics version of WACCM and observations from various satellite and ground based platforms. The study compares a simulation forced with observed "real world" boundary conditions of the major ozone depleting substances (ODS), with that forced with a "world avoided" scenario in which ODSs increase 3.5%/year after 1985. The paper demonstrates that under the extreme meteorological conditions of spring 2020, significantly greater Arctic ozone depletion would have occurred were it not for the Montreal Protocol. The paper also performs several sensitivity simulations to assess the model denitrification compared with MLS data, and refines the assumptions of the model NAT aerosol density.

This is a well written paper that presents some important results concerning the stratospheric ozone impacts as a result of the Montreal Protocol. It follows other recent "world avoided" studies, focusing on the very cold Arctic conditions during spring 2020. I found the analysis to be mostly clear and the figures well presented. I have only a few minor comments and a suggestion (listed below) which should be mostly straightforward to address, but otherwise recommend publication of the manuscript.

Comments/corrections:

1) Figure 2a: This is a nice figure, but please try to enlarge the map in the lower left corner, or at least make the lettering significantly larger. The location names are unreadable as is (at least in the version I have).

The map in Figure 2a has been redone with darker lines, larger markers, and no text labels. We believe this makes reference to the markers in the legend of the whole figure easier, as the lettering is no longer competing with the geographic lines in the map.

2) L91-92: "… assumes a 3.5% per year increase …" – Please provide a little more explanation on how this increase was determined.

We agree that this assumption needs further justification and have added the following explanation in Lines 75-78:

"The assumption of 3.5% per year growth matches that used in the Garcia et al. (2012) World Avoided study and is a good approximation of the grown rates seen in years immediately prior to emissions controls, thus representing an illustrative "business as usual" alternate history."

3) L162: "Although temperature histories can also be important …." – "temperature histories" should be briefly defined/explained. I assume the authors are referring to how much the back

trajectories of parcels encounter (or not) temperatures cold enough for PSC formation and ozone loss.

 We agree that this term should be defined. We have elaborated on this point and reorganized the text as follows in Lines 182-187:

"The figure shows that the largest depletion here tracks the lowest local temperatures of the profile (temperature shown in solid red). Although temperature histories can also be important, as activation can persist in air parcels which previously encountered cold air but are currently above the temperature threshold for PSC formation, this broadly supports the view that much of this year's ozone loss was related to widespread local cold temperatures increasing the efficiency of heterogeneous reactions on PSCs (Wohltmann et al., 2020; Manney et al., 2020)."

4) L167: should read: "… the Microwave Limb Sounder (MLS) instrument onboard the Aura satellite …"

 This has been corrected.

5) L175: "Figure 3 shows the progression of six increasingly denitrified ...." – only four model panels (a-d) are shown (not six). This is also referred to on L181-182.

 These references have been corrected and we have reworked both the explanation of Figure 3 and its figure caption to make it clearer which panel is being referred to when.

6) L188: "dailyminimum" – separate into two words.

This has been orrected.

7) L298, citation should read: "Langematz, U. and M., Tully (Lead Authors), Calvo, N., Dameris, M., ....."

This has been corrected.

8) Given that the WA vs. RW ozone difference in spring 2020 is especially large (e.g., Fig. 4 shows ~100DU difference persisting into late April 2020), and larger than the 2010-2020 SH spring (late Oct - Dec in Fig. S4), it would be useful to show the WA vs. RW UV index, or at least the ratio in surface UV flux (WA/RW) vs. wavelength for the Arctic late spring or early summer 2020 (e.g., 70N - 90N avg).

The change in surface UV was shown in the previous WA studies (e.g., Newman et al., 2009, Garcia et al., 2012) as an important end result of ozone depletion. The authors briefly mention this point in the Conclusions, but showing a figure and briefly quantifying the resulting change in surface UV in the Arctic late spring/summer would be useful to include in the present paper. However, I'll leave it to the authors whether or not to pursue this depending on the logistical and time constraints involved (e.g., re-running the model to get the needed output, etc.).

 We agree that further investigation and quantification of surface UV changes would be interesting to look at. While the information is not an SD-WACCM model output, we estimated

the change in the clear-sky UV index at noon for four northern cities using the SD-WACCM total column ozone from our simulations and tabulated the results in the new Table 1. They are described in the text in Lines 276-282:

"A coupled biosphere model would be required to fully investigate such effects, but we can estimate the change in surface UV during late March and April 2020 by calculating the clear-sky UV Index at noon for the SD-WACCM grid-point nearest four northern cities using the procedure outlined in Burrows et al. (1994), with the results shown in Table 1. While the UV Indices at the end of March in the WA run show substantial percentage increases (from 25% to 88%), the absolute values are still quite small due to the large zenith angles in the Arctic spring. By the end of April, however, there are still substantial differences between the runs, despite active ozone depletion having ceased for the year, reinforcing that there are increasing year-round impacts, as seen in Figure 7."

**Citation**: https://doi.org/10.5194/acp-2020-1297-RC2

---

## Author Response (AR2)

**Dear Editor:**

We would like to thank the editor for the comments and suggestions for improvement. Below please find a point-by-point explanation of how we have modified our submitted manuscript.

1. Abstract line 7. Before giving the specific details of the modelled World Avoided simulation it is necessary to say what hypothetical scenario it is based on. Otherwise the numbers will not have any context. The abstract needs a second sentence which give the assumed halocarbon scenario.

We agree that adding a sentence to the abstract with the assumed halocarbon scenario provides essential context. The beginning of the abstract now reads (changes in **bold**):

"Without the Montreal Protocol the already extreme Arctic ozone losses in boreal spring of 2020 would be expected to have produced an Antarctic-like ozone hole, based upon simulations performed using the Specified Dynamics version of the Whole Atmosphere Community Climate Model (SD-WACCM) **using an alternate emission scenario of 3.5% growth in ozone depleting substances from 1985 onwards.** In particular, **we find that** the area of total ozone below 220 DU, a standard metric of Antarctic ozone hole size, would have covered about 20 million km2. Record observed local lows..."

2. I have seen that in the GRL/JGR Arctic Ozone special issue there is an update to the cited Chipperfield et al (2015) study of winter 2011 (Feng et al 2021), which uses the same model to look at winter 2020 (in the Supplementary Material). The availability of this should be mentioned around line 62.

https://agupubs.onlinelibrary.wiley.com/doi/full/10.1029/2020GL091911

We thank the editor for pointing out the availability of this study, and have added the citation and changed the relevant sentence to the following:

"Chipperfield et al. (2015) examined the WA for the recent extreme cold year of 2011. They found that Arctic ozone levels would indeed have dropped below 220 DU in that year in the WA, but in a limited region that did not span the entire pole as in the Antarctic, as we discuss in Section 3. During a recent study quantifying drivers of depletion in Spring 2020 in the real world (Feng et al., 2021), they updated their previous WA model and found that the year 2020 would have seen much deeper column depletion during March than 2011 did (see their Supplementary Information)."

3. Line 260-261. 'We have ... ecosystem'. I have a problem with the logic of this sentence. It is based on an assumed ODS scenario and also assumes that the meteorology of all the previous years would have been the same as actually occurred, even with a different atmospheric

composition. Maybe a cold year would have happened in 2019, for example, in an actual noprotocol world. I think what you can say is the 2020 meteorology would have caused an 'Arctic ozone hole' in your assumed scenario. I think you should phrase the message I this way.

We agree that the assumption of 2020 meteorology may not have remained the same in a noprotocol world, and have rewritten the sentence to clarify this as follows:

"We have demonstrated that, were it not for the Montreal Protocol, the meteorological conditions seen in 2020 would have produced the first Antarctic-like ozone hole over the Arctic, an area with a substantial human population and vibrant ecosystem."